# Forward-predictive SERS-based chemical taxonomy for untargeted structural elucidation of epimeric cerebrosides

Emily Xi Tan [1], Shi Xuan Leong[1], Wei An Liew[1], In Yee Phang[2], Jie Ying Ng [3], Nguan Soon Tan [4,5], Yie Hou Lee [3,6,7] ✉ & Xing Yi Ling [1,2,4,8] ✉

Achieving untargeted chemical identification, isomeric differentiation, and quantification is critical to most scientific and technological problems but remains challenging. Here, we demonstrate an integrated SERS-based chemical taxonomy machine learning framework for untargeted structural elucidation of 11 epimeric cerebrosides, attaining >90% accuracy and robust single epimer and multiplex quantification with <10% errors. First, we utilize 4-mercaptophenylboronic acid to selectively capture the epimers at molecular sites of isomerism to form epimer-specific SERS fingerprints. Corroborating with *in-silico* experiments, we establish five spectral features, each corresponding to a structural characteristic: (1) presence/absence of epimers, (2) monosaccharide/cerebroside, (3) saturated/unsaturated cerebroside, (4) glucosyl/galactosyl, and (5) GlcCer or GalCer's carbon chain lengths. Leveraging these insights, we create a fully generalizable framework to identify and quantify cerebrosides at concentrations between $10^{-4}$ to $10^{-10}$ M and achieve multiplex quantification of binary mixtures containing biomarkers $GlcCer_{24:1}$, and $GalCer_{24:1}$ using their untrained spectra in the models.

Achieving untargeted structural elucidation, isomeric differentiation, and quantification is a paramount goal in molecular characterization and crucial to resolving many scientific and technological problems[1–3]. Surface-enhanced Raman scattering (SERS) spectroscopy emerges as a compelling qualitative and quantitative molecular sensing approach because it rapidly provides rich vibrational information for univocal chemical identification and multiplexing capabilities at ppm to ppb levels[4–6]. Currently, identifying known chemical molecules through manual referencing of existing SERS databases and literature is relatively straightforward, However, manually matching SERS peaks to

vibrational modes is tedious and error-prone, especially when handling complicated spectra and large datasets. Moreover, it is a passive approach and lacks the forward-inferring capabilities to predict "unidentified molecules" beyond the boundaries of existing databases[4–9]. The challenge lies in the untargeted identification of "unidentified molecules" outside existing SERS databases.

Inspired by taxonomy, based on the use of anatomical and behavioral characteristics to classify new species, we postulate the establishment of a SERS-based chemical taxonomy that can achieve untargeted identification of "unidentified molecules" by combining

[1]School of Chemistry, Chemical Engineering and Biotechnology, Nanyang Technological University, 21 Nanyang Link, Nanyang 637371, Singapore. [2]School of Chemical and Material Engineering, Jiangnan University, Wuxi 214122, People's Republic of China. [3]KK Research Centre, KKH, 100 Bukit Timah Road, Singapore 229899, Singapore. [4]Lee Kong Chian School of Medicine, Nanyang Technological University, 59 Nanyang Drive, Singapore 636921, Singapore. [5]School of Biological Sciences, Nanyang Technological University Singapore, 60 Nanyang Drive, Singapore 637551, Singapore. [6]Obstetrics and Gynaecology Academic Clinical Program, Duke-NUS Medical School, Singapore 169857, Singapore. [7]Critical Analytics in Manufacturing Personalized Medicine, Singapore-MIT Alliance for Research and Technology, 1 CREATE Way, #04-13/14 Enterprise Wing, Singapore 138602, Singapore. [8]Institute for Digital Molecular Analytics and Science (IDMxS), Nanyang Technological University, 59 Nanyang Drive, Singapore 636921, Singapore. ✉e-mail: yiehou.lee@smart.mit.edu; xyling@ntu.edu.sg

SERS fingerprints with a hierarchical machine learning (ML) framework[10–12]. To begin, we establish hierarchical levels within the SERS-based chemical taxonomy. Each level is linked to a molecular structural characteristic, such as the types and numbers of functional groups[10,13]. Leveraging the taxonomic ML model, we can predict individual structural attributes in a stepwise manner. This progressive process can be done by analyzing and pairwise profiling similarities and differences in structure and SERS spectra. Crucially, this approach facilitates unprecedented forward prediction, allowing for the deduction of "unidentified molecules" situated beyond the boundaries of the ML model. Specifically, our proposed process systematically excludes alternative structural possibilities when the SERS spectra traverse the hierarchical levels of the chemical taxonomy, culminating in the precise identification of the exact molecular structure. In contrast, such forward prediction remains elusive through a single classification ML model, which inaccurately classifies the "unidentified molecules" as one of the pre-existing labeled classes in that model.

One biomolecule class that directly benefits from such a SERS chemical taxonomy model is the cerebrosides. Particularly, the epimeric glucocerebrosides (GlcCer$_{X:Y}$) and galactocerebrosides (GalCer$_{X:Y}$) differ in the spatial orientation of their $C_4$ OH-groups ($C_4$ site of isomerism) in their glycosyl/ galactosyl moiety and consist of ceramides moieties with varied carbon chain length (X) and saturation degrees (Y)[2,14]. Due to their structural diversity, they possess different bioactivities and play distinct functional and constitutional roles in cellular signaling and metabolism. For instance, an increase in GlcCer$_{24:1}$ alludes to endometriosis and Gaucher disease, whereas GalCer$_{24:1}$ is implicated in Fabry and Krabbe diseases[2,14–19]. At present, a rapid, point-of-care platform for their untargeted identification is pertinent as isomeric differentiation and quantification using gold standard gas/liquid chromatography-mass spectrometry remain arduous due to fragmentation pattern issues and inefficient prior derivatization[2].

Herein, we establish a SERS-based chemical taxonomy using hierarchical ML with forward-inferring capabilities for untargeted structural elucidation of eleven (11) GlcCers and GalCers, attaining classification accuracy >90% through their SERS spectra that are untrained within the model. Our SERS-ML framework is proficient in single and multiplex quantification, achieving precision with <10% errors at their physiological relevant concentrations (Fig. 1). To achieve this SERS-based chemical taxonomy, we first develop an Ag SERS platform functionalized with 4-mercaptophenylboronic acid (4-MPBA) to specifically capture the epimers at their $C_4$ site of isomerism, yielding unique epimer-MPBA adducts, each with distinct SERS fingerprints. (Fig. 1a, b) Corroborating the SERS fingerprints with DFT simulations allows us to identify five key spectral features, each corresponding to a key structural characteristic, i.e., (1) the presence or absence of epimers, (2) monosaccharide vs. cerebroside, (3) saturated vs. unsaturated ceramide, (4) glucosyl vs. galactosyl moieties, and (5) GlcCer or GalCer's carbon chain lengths. We then perform feature engineering of the SERS spectra to extract individual peak spectral features, such as position, intensity, full width at half maximum, skew, and ratio, as ML inputs for accurate and efficient modeling (Fig. 1c)[20]. Using spectral features as ML inputs, we build a hierarchical ML framework consisting of four sequential random forest classifiers (RF-C1–4) and two support vector machine regressors (SVM-R 5.1 and 5.2) (Fig. 1d). This framework elucidates the five identified structural characteristics from each model and then aggregates the information gained to reconstruct the complete molecular structure. Collectively, the four RF-Cs yield classification accuracies surpassing 90%, whereas the two SVM-Rs accurately predict the carbon chain lengths up to 1 carbon difference, all when using untrained spectra for blind tests, highlighting the generalizability of our framework for structural elucidation of all 11 cerebrosides. Importantly, we prove that although the model is established using spectra of cerebrosides at $10^{-4}$ M, it is still effective in predicting "unidentified cerebrosides" at concentrations

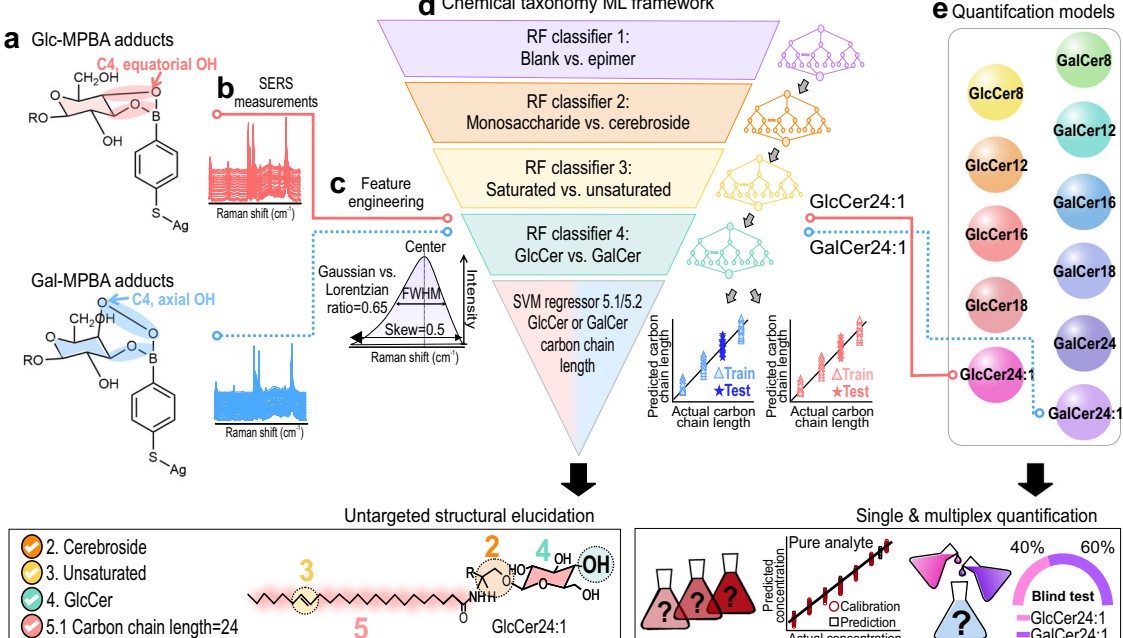

**Fig. 1 | Schematics of a forward-predictive integrated SERS-based chemical taxonomy machine learning framework. a** The SERS probe 4-MPBA first covalently captures the epimeric cerebrosides in unique configurations to form specific cerebroside-MPBA complexes. **b** The unique cerebroside-MPBA complexes will have distinctive SERS spectra. **c** We perform feature engineering to extract SERS spectral features from each SERS peak to form the machine learning input. **d** To perform untargeted structural elucidation, the spectra features of an "unidentified" epimeric cerebrosides at unknown trace concentrations is fed into our chemical taxonomy which will elucidate the chemical structure and nomenclature. **e** Next, the spectra features are parsed into our pretrained quantification models for single and multiplex quantification. Glc-MPBA glucosyl-MPBA, Gal-MPBA galactosyl-MPBA, GlcCer glucocerebroside, GalCer galactocerebroside, RF random forest, SVM support vector machine.

1–6 orders of magnitude lower than those in the trained model (i.e., at $10^{-5}$–$10^{-10}$ M) with accuracies ranging from 87 to 100% with <1 carbon chain length discrepancy. This demonstrates the robustness and applicability of our chemical taxonomy framework for practical SERS sensing applications, where the concentration of analytes is frequently unknown. Furthermore, our integrated ML framework allows seamless identification and quantification with <10% errors for all 11 cerebrosides from $10^{-4}$ to $10^{-10}$ M (Fig. 1e). We further achieve multiplex quantification of biomarkers $GlcCer_{24:1}$ and $GalCer_{24:1}$ in binary mixtures at µM range with an absolute error of <4% between predicted and actual concentrations in 30 blind samples. Overall, our forward-predictive SERS-based chemical taxonomy marks a pivotal advance from existing molecular identification methods, realizing the long-standing goal of rapid (<30 min), untargeted structural elucidation and quantification. In this work, we create a localized SERS molecular space, within which our ML framework can both interpolatively and extrapolatively predict 11 gluco- and galactocerebrosides. We envision high-throughput testing of more probes and analyte combinations to further extend the framework and create a global SERS molecular space capable of elucidating other classes of isomeric compounds to meet escalating demands for rapid, point-of-need analytical tools.

## Results and Discussion

### SERS platform and biomolecule characterization

To establish a reliable SERS-based chemical taxonomy ML framework for untargeted structural elucidation of unknown epimeric cerebrosides, it is pertinent to generate a SERS database with distinctive and strong SERS signals to facilitate their unambiguous identification and differentiation. We utilize a "capture and confine" strategy to first covalently capture the epimers onto 4-MPBA grafted on Ag nanocubes before physically concentrating the mixture using a hydrophobic perfluorothiol-Ag substrate to amplify SERS signals (analytical enhancement factor = $3.2 \times 10^5$) (Suppl. Notes 1–4). We employ this indirect SERS detection due to the small Raman cross sections of the epimers. The MPBA probe effectively forms covalent boronate ester bonds with epimers' 1,2-diol group directly at their $C_4$ site of isomerism, which generates distinct SERS fingerprints from each unique epimer-MPBA adduct, facilitating their differentiation (Fig. 2a, b)[9,21,22]. In our investigation, we study 11 cerebrosides, including five gluco-cerebrosides ($GlcCer_{X:Y}$), and six galactocerebrosides ($GalCer_{X:Y}$) at $10^{-4}$ M, where X represents ceramide chain length and Y denotes saturation degrees. We further compare them to glucose and galactose which are identified as primary interferences since they have the exact same hexose moiety and are expected to bind in similar orientations/configurations to MPBA as gluco- and galactocerebrosides. We categorize cerebrosides according to five specific structural characteristics. In category 1 (blank vs. epimers), we first determine the presence or absence of epimers. (Fig. 2c). In category 2 (monosaccharide vs. cerebroside), we differentiate monosaccharides from cerebrosides, which have ceramide chains glycosidically linked to the glycosyl moiety. In category 3 (saturated vs. unsaturated), the cerebrosides are distinguished by the ceramide's saturation degrees (Y = 0 or 1). In category 4 (glucosyl vs. galactosyl), the epimeric cerebrosides are differentiated based on the presence of either a glycosyl ($C_4$ equatorial OH) or galactosyl ($C_4$ axial OH) moiety. Finally, in category 5 (GlcCer or GalCer carbon chain length), the precise ceramide carbon chain length is identified. In the experimental epimer-MPBA adduct SERS spectra, we note overall varied degrees of red/blueshifts and intensity changes at five regions due to differential covalent interactions between MPBA and epimers (Fig. 2d).

### Elucidating epimer-specific SERS fingerprints

To confirm the chemical relevance of spectral features driving the differentiation among the 11 cerebrosides, we further scrutinize the high-veracity SERS fingerprints and corroborate them with density

functional theory (DFT) simulations to elucidate the molecular origins of various spectral variations. We identify five vital spectral regions from the spectra relating to the epimers' structural characteristics (Fig. 3a). They include (1) the presence or absence of epimers at 1330 $cm^{-1}$, (2) monosaccharide vs. cerebroside at 1300 $cm^{-1}$, (3) saturated vs. unsaturated ceramide at 1595–1603 $cm^{-1}$, (4) glucosyl vs. galactosyl at 414–419 $cm^{-1}$, and (5) ceramide carbon chain lengths of GlcCer at 1023 $cm^{-1}$ or GalCer at 687 $cm^{-1}$. First, to confirm that the SERS platform has successfully captured the epimers (category 1), we note a sharp decrease of the BOH bending ($\nu CH + \beta CH + \beta BOH$) at 1330 $cm^{-1}$ from 0.6 to <0.2 arbitrary units (arb. u.) for all epimers (Fig. 3b), indicating the formation of boronate ester bonds with their 1,2-diol moiety via a condensation reaction[4,21,22]. Importantly, we can utilize this mode to differentiate between monosaccharides-MPBA ($I_{1330}/I_{1567} < 0.07$ arb. u.) and cerebrosides-MPBA (0.07 to 2.2 arb. u.), whereby their difference in peak intensity ratio is statistically significant ($p < 0.05$, category 2, Fig. 3b). For monosaccharides, the 1330 $cm^{-1}$ peak intensity is significantly lower because they have (1) more OH groups available per molecule to interact with MPBA and (2) more compact molecular structures with no steric hindrance from bulky side chains, which increases accessibility to MPBA for preferential binding. Next, we note that the differentiation between the saturation vs. unsaturation in the ceramide (category 3) is correlated to the totally symmetric $\nu CC$ stretching mode at 1584–1601 $cm^{-1}$ (Fig. 3c). Compared to MPBA blanks' $\nu CC$ mode at 1603 $cm^{-1}$, saturated $GalCer_{24}$ slightly redshifts to 1601 $cm^{-1}$, whereas the unsaturated $GalCer_{24:1}$ and $GlcCer_{24:1}$ experience an increase in intensity and more pronounced redshifts to 1595 and 1591 $cm^{-1}$, with respect to the non-totally symmetric $\nu CC$ peak at 1575 $cm^{-1}$. This trend agrees well with DFT, where we notice an increase in intensity and redshift of the same SERS band from blank at 1621–1616 $cm^{-1}$ for saturated $GalCer_{24}$ and 1607 $cm^{-1}$ for unsaturated $GalCer_{24:1}$ and $GlcCer_{24:1}$ with respect to the non-totally symmetric $\nu CC$ peak at 1525 $cm^{-1}$. This is attributed to the presence of a distal C = C in unsaturated $GalCer_{24:1}$ and $GlcCer_{24:1}$, which is capable of forming $\pi$-$\pi$ interactions with MPBA's benzene ring and disrupting MPBA's molecular symmetry, resulting in redshifting and intensity increase in the totally symmetric $\nu CC$ mode due to Herzberg–Teller contribution[23,24]. In category 4, GlcCer and GalCer can be differentiated using the peak position of the $\beta CCC + \nu CS$ band in the range of 414–419 $cm^{-1}$. Compared to MPBA-blanks at 414 $cm^{-1}$, GlcCer-MPBA adducts blueshift slightly to between 414 and 417 $cm^{-1}$, whereas GalCer-MPBA undergoes significant blueshifts to 417–419 $cm^{-1}$ (Fig. 3d). This is in good accordance with our simulated spectra, where the same $\beta CCC + \nu CS$ vibrational mode blueshift more for GalCer-MPBA than GlcCer-MPBA, from 488 $cm^{-1}$ in blank-MPBA to 593–595 $cm^{-1}$ for the former and to 507–509 $cm^{-1}$ for the latter. This is because GalCer-MPBAs' 5-membered ring formed is less strained due to the axial OH group, with bond angles of 102.8°–106.2° that are overall closer to the ideal 107° for five-membered rings compared to GlcCer-MPBAs' 102.7°–105.4°, which collectively experience more ring strain (Suppl. Note 5). A decrease in ring strain leads to a more substantial induction effect, which in turn leads to an increase in both benzene electron cloud delocalization and polarizability of the C–S bond, resulting in a larger blueshift for GalCer-MPBA[24,25]. Finally, for category 5, C–H bending ($\beta CH$) mode at 1023 $cm^{-1}$ shows strong positive correlations and hence is used to elucidate the ceramide carbon chain length for Glc-MPBA adducts (Fig. 3e). Comparing the $I_{1023}/I_{1567}$ peak ascribed to $\beta CH$ of MPBA's benzene against increasing carbon chain length, we observe positive correlations in the experimental and our DFT-simulated SERS spectra. The increase in $\beta CH$ intensity is likely due to the more significant extent of symmetry breaking of the 4-MPBA from nearly $C_{2v}$ to $C_s$ after binding to the various Glc-MPBA with increasing carbon chain lengths[9,21,24]. The non-ideal bond angles of Gal-MPBA adducts likely aggravate this symmetry-breaking effect. Similarly, for Gal-MPBA, the $I_{691}/I_{1567}$ peak indexed to the $\beta CCC + \nu CS$ mode

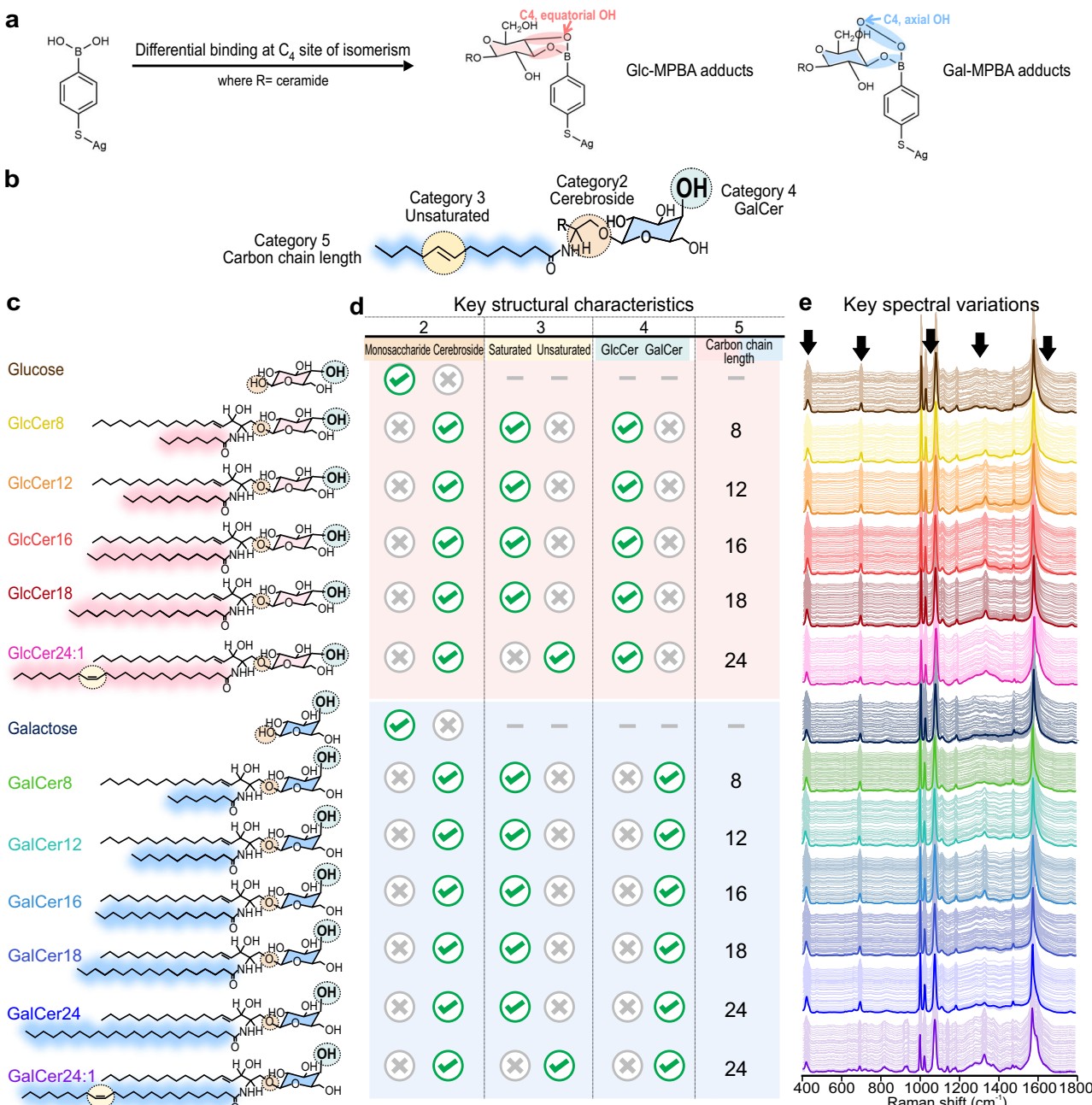

**Fig. 2 | Key structural characteristics of 13 glucosyl(glc) and galactosyl(gal)-analytes enabling their SERS differentiation. a** Molecular structures of 4-mercaptophenylboronic acid (4-MPBA) before and after differential covalently bonding to C4 of glucosyl(glc) and galactosyl(gal)-analytes. **b** The general structure of cerebrosides with various categorized structural characteristics. **c** Molecular structures of the 13 epimers, **d** Their key structural characteristics, and **e** their corresponding differential normalized SERS spectra.

positively correlates to increasing carbon chain length, which agrees with DFT spectral changes (Fig. 3f). Notably, this increase in intensity is prominent in Gal-MPBA adducts due to the aforementioned ideal bond angles effectuating strong induction effect, causing a concomitant increase in benzene electron cloud delocalization and C–S bond polarization for Gal-MPBA with longer carbon chain lengths[9,21,24,25]. The differential peak intensities for various epimers thus reflect their carbon chain length specific differences and underscore the high propensity for quantitative chemical structure-spectra correlations. These five unique structural characteristics captured in the SERS fingerprints are evidence of 13 epimers' differential covalent interactions with MPBA, which paves the way for their unambiguous identification and differentiation using ML.

## Supervised and unsupervised ML for structural elucidation

Leveraging the strong correlations between the cerebrosides' SERS fingerprints and their structural characteristics, we create a universal SERS-based chemical taxonomy, achieving over >90% classification accuracy and <1 carbon chain length difference. This employs ML's advanced capabilities to discern underlying data patterns, enabling instantaneous, untargeted structural elucidation across any concentration[20,26–29]. To enhance ML accuracy and efficiency by reducing modeling time, we parameterize spectra and isolate 19 peaks from individual cerebroside spectra and derive five peak attributes, including position, intensity, full width at half maximum, skew (symmetry or degree of asymmetry), and ratio (degree of Gaussian/Lorenztian characteristics). This effectively reduces the input features from 1200

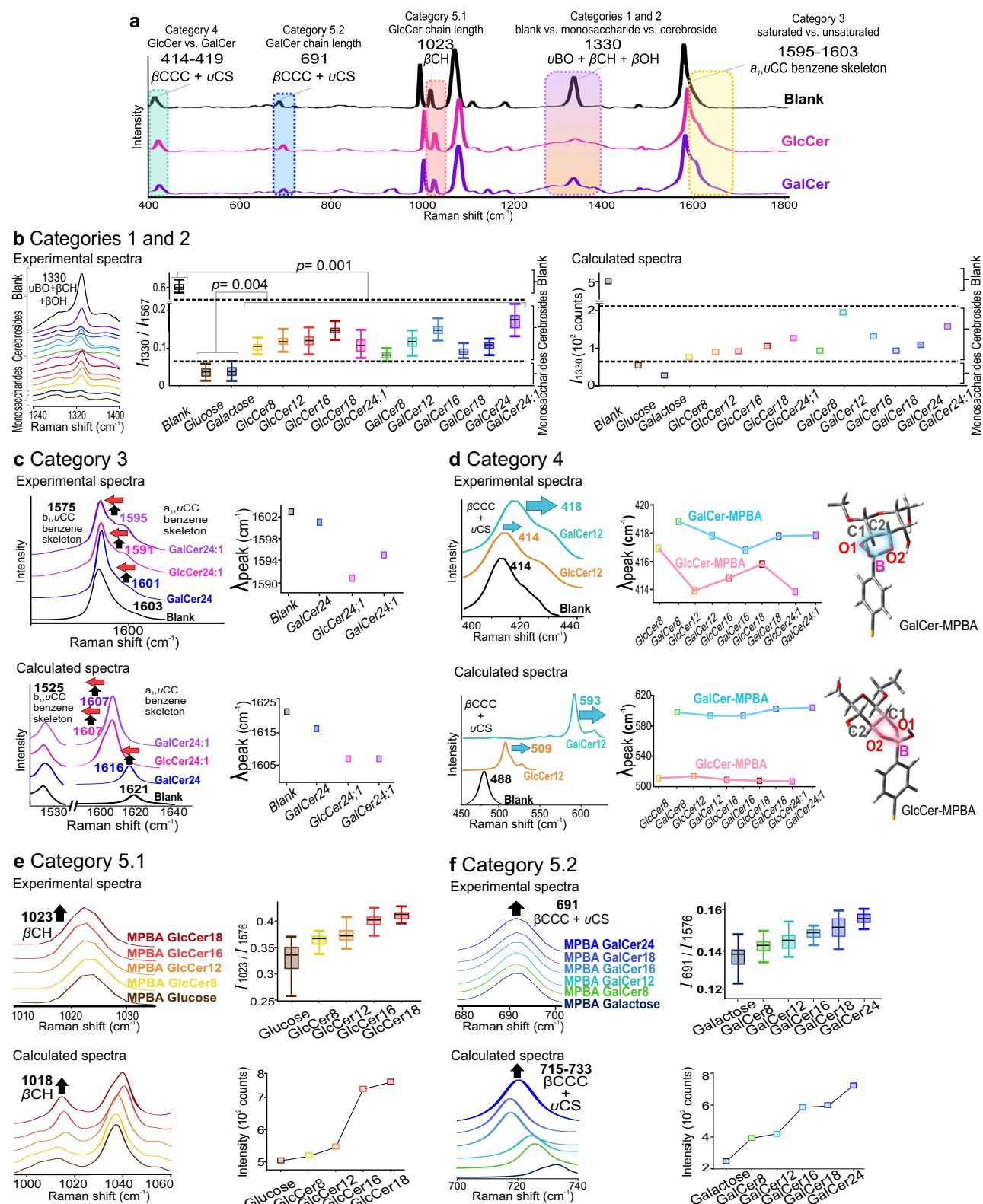

variables in a single SERS spectrum to just 19 × 5 = 95 features for ML (Suppl. Notes 6). To begin, we use unsupervised t-distributed stochastic neighbor embeddings (t-SNE) clustering to confirm that SERS fingerprints can differentiate all epimers (Fig. 4a). t-SNE primarily serves to visualize high-dimensional data by projecting it into two-dimensional space. From this t-SNE analysis, we observe distinct clusters of cerebrosides, indicating that the cumulative differences encoded in the spectra are significant (Fig. 4a)[29]. Cerebrosides are

distinguished from the interfering monosaccharides (glucose and galactose). Individual cerebrosides also cluster according to their structural characteristics in our SERS molecular space. Moreover, we note clear segregations in the subsequent four t-SNE plots, which were segmented based on their structural characteristics into (1) the presence vs. absence of epimers (2) monosaccharide vs. cerebroside, (3) saturated vs. unsaturated ceramide and (4) GlcCer vs. GalCer (Fig. 4b−e). Overall, our unsupervised t-SNE results provide

**Fig. 3 | SERS analysis of 11 cerebrosides and 2 monosaccharides.**
**a** Representative SERS spectrum of 4-MPBA, $GlcCer_{24:1}$, and $GalCer_{24:1}$. Key SERS regions corresponding to the structural characteristics 1−5, where $\upsilon$ = stretching, $\beta$ = bending, $\gamma$ = wagging, $a_1$ = totally symmetric, $b_2$ = non-totally symmetric.
**b** Categories 1 and 2 compare the experimental and calculated spectra of MPBA, monosaccharides-MPBA, and cerebroside-MPBA adducts at 1330 cm$^{-1}$ ascribed to $\beta BO + \beta CH + \beta OH$. The peak intensity ratio difference for each of the 13 analytes is extracted from 60 individual SERS spectra and plotted in standard boxplots with interquartile ranges shaded, mean values indicated, and the whiskers indicating min and max, respectively. **c** Category 3 is a comparison of the experimental and calculated spectra as well as the intensity ratio difference of the $a_1,\upsilon CC$ mode between saturated vs. unsaturated cerebroside-MPBA, which shows differential degrees of redshifting compared to MPBA. **d** Category 4, comparison of the experimental and calculated $\beta ccc + \upsilon CS$ peak position difference between epimeric GlcCer-MPBA and GalCer-MPBA adducts. GlcCer-MPBA adducts with bond angles

of the five-membered ring between 102.7° and 105.4° experience higher collective ring strain compared to GalCer-MPBA adducts with bond angles between 102.8° and 106.2° which are closer to the ideal 107°. **e** Category 5.1 elucidates Glc-MPBA carbon chain length effects by comparing the experimental and calculated peak intensity ratio of $\beta CH$ of the various Glc-MPBA with different chain lengths. The peak intensity ratio difference for each of the five analytes is extracted from 60 individual SERS spectra and plotted in standard boxplots with interquartile ranges shaded, mean values indicated, and the whiskers indicating min and max, respectively. **f** Category 5.2 elucidates Gal-MPBA carbon chain length effects by comparing the experimental and calculated peak intensity ratio of $\beta ccc + \upsilon CS$ of the various Gal-MPBA with different chain lengths. The peak intensity ratio difference for each of the six analytes is extracted from 60 individual SERS spectra and plotted in standard boxplots with interquartile ranges shaded, mean values indicated, and the whiskers indicating min and max, respectively.

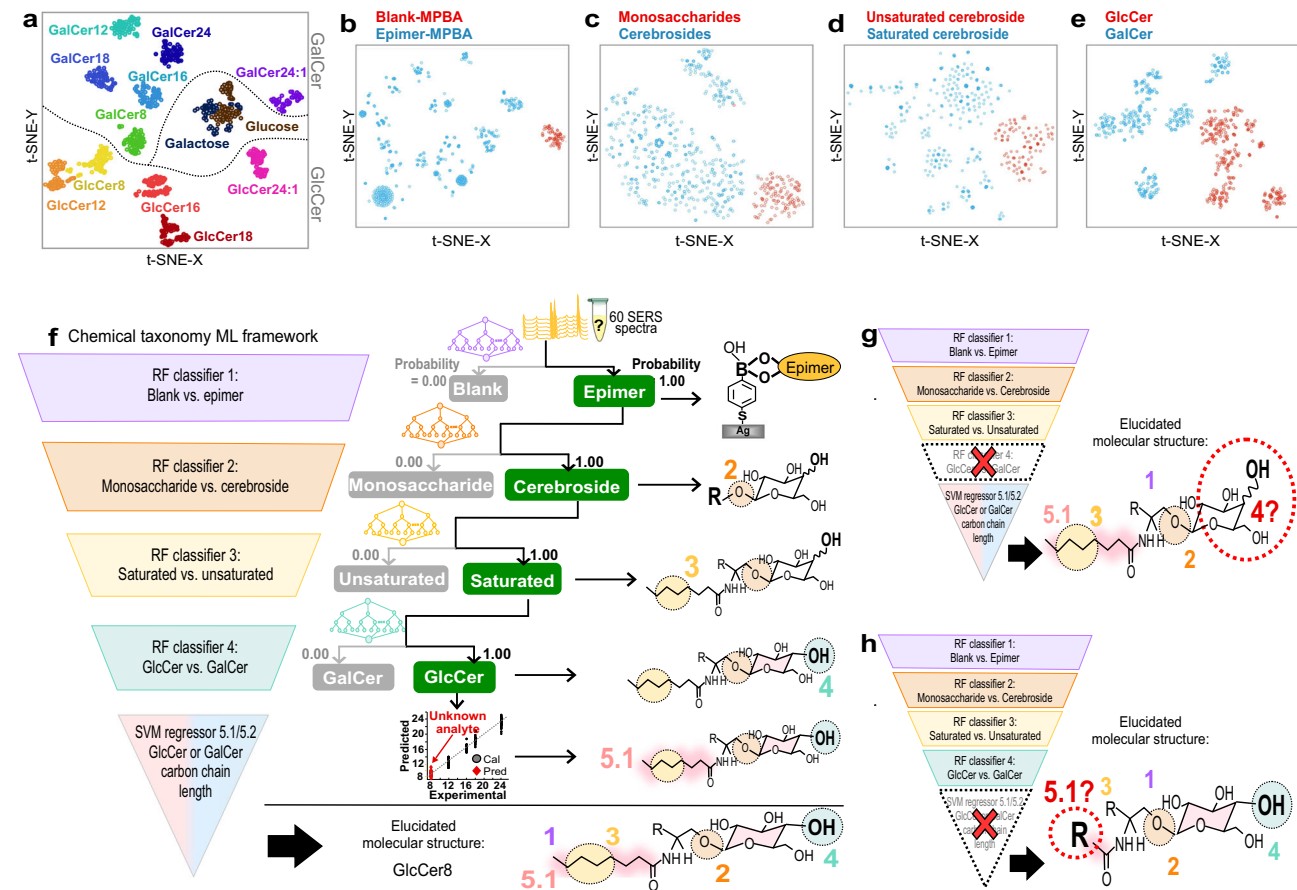

**Fig. 4 | Unsupervised and supervised machine learning results to forward-predict "unidentified" cerebrosides. a** Visualization of the cerebroside SERS molecular space using unsupervised t-distributed stochastic neighbor embedding (t-SNE) showing distinct clustering in **b**. blank-MPBA vs. epimer-MPBA, **c** monosaccharides-MPBA vs. cerebrosides-MPBA, **d** unsaturated vs. saturated

cerebrosides-MPBA, and **e** GlcCer-MPBA vs. GalCer-MPBA. **f** Forward prediction of "unidentified" cerebroside structures using the five-level SERS-based chemical taxonomy framework consisting of a hierarchical ensemble of six ML models. Contrasting incomplete structural elucidation when **g** RF-C4 and **h** SVM-R 5 are removed respectively.

unambiguous differentiation among the various epimers without human input. This sets the stage for untargeted structural elucidation using supervised ML models.

A five-level chemical taxonomy model is at the core of our forward-inferring ML framework for untargeted elucidation. It comprises four sequential random forest classifiers (RF-C1−4) to determine a specific structural characteristic and two support vector machine regressors (SVM-R 5.1 and 5.2) to estimate GlcCer and GalCer's carbon chain lengths. (Fig. 4f)[26]. We chose the RF-C because tree-based algorithms are robust for deciphering both linear and non-linear relationships amongst variables and data patterns that are challenging by

manual analysis[30]. The SVM-R excels in handling complex boundary-specific problems. In our case, the goal is to find the hyperplane that best segregates GlcCer and GalCer chain lengths in the dataset[28]. The hierarchical ML-driven framework is designed for progressive structural elucidation of cerebrosides based on their five-tiered structural characteristics, akin to biological taxonomic analysis. After confirming the presence of epimers, the framework sequentially predicts the following structural characteristics: (category 2) monosaccharide vs. cerebroside, (3) saturated vs. unsaturated ceramide, (4) glucosyl vs. galactosyl moieties, and (5.1 or 5.2) GlcCer or GalCer's carbon chain lengths. We emphasize that a sequential approach is necessary for

**Table 1 | Fully generalizable forward prediction of all 11 cerebrosides when the respective dataset of the individual cerebrosides is excluded in the training set and used as blind tests**

| Cerebroside used as blind test | RF-C1 (Blank vs. epimer) | RF-C 2 (Monosaccharide vs. cerebroside) | RF-C 3 (Saturated vs. unsaturated) | RF-C4 (GlcCer vs. GalCer) | Overall RF-C1 to C4 | SVM-R 5.1 & 5.2 (Carbon chain length) | |
|---|---|---|---|---|---|---|---|
| | Classification accuracy | Classification accuracy | Classification accuracy | Classification accuracy | Overall classification accuracy | Predicted chain length | Absolute difference |
| GlcCer8 | 100% | 100% | 100% | 95% | 95% | 9 ± 1 | 1 |
| GlcCer12 | 100% | 100% | 100% | 92% | 92% | 12 ± 1 | 0 |
| GlcCe16 | 100% | 100% | 100% | 100% | 100% | 16 ± 1 | 0 |
| GlcCer18 | 100% | 100% | 100% | 90% | 90% | 18 ± 1 | 0 |
| GlcCer24:1 | 100% | 93% | 90% | 100% | 84% | - | -- |
| GalCer8 | 100% | 100% | 100% | 100% | 100% | 9 ± 1 | 1 |
| GalCer12 | 100% | 100% | 100% | 100% | 100% | 13 ± 2 | 1 |
| GalCer16 | 100% | 100% | 100% | 100% | 100% | 17 ± 1 | 1 |
| GalCer18 | 100% | 100% | 100% | 100% | 100% | 18 ± 1 | 0 |
| GalCer24 | 100% | 100% | 100% | 100% | 100% | 24 ± 1 | 0 |
| GalCer24:1 | 100% | 100% | 100% | 100% | 100% | - | - |
| Average | 100 ± 0% | 99 ± 2% | 99 ± 3% | 98 ± 4 % | 97 ± 6% | 0.4 ± 0.5% | |

untargeted identification. It systematically reconstructs individual structural characteristics while eliminating other structural possibilities to elucidate exact molecular structure and identity. This is impossible with the commonly employed single ML classification model, which would erroneously force unknown samples into existing labeled classes.

Using $GlcCer_8$ samples for blind testing, their SERS spectra undergo evaluation within the chemical taxonomy framework. Our RF-C1 correctly identifies each spectrum as an epimer-MPBA, not as unreacted MPBA with 100% certainty (probability = 1) (Fig. 4f). Next, the spectra are directed to RF-C 2, where they are compared with either monosaccharide-MPBA or cerebroside-MPBA groups. They are then classified as belonging to the biomolecule class of cerebrosides. The spectra then proceed to RF-C 3, which predicts whether they are saturated or unsaturated cerebrosides (Y = 0 or 1). Our RF-C 3 precisely predicts they are saturated cerebrosides. In the final classifier, RF-4 accurately predicts that the spectra belong to GlcCer-MPBA. After determining the structural characteristics using these four different classifiers, the spectra are channeled to the regressor (SVM-R 5.1 for GlcCer), which determines that the carbon chain length of the bonded ceramide moiety is 8. The overall probability scores for all blind test spectra from each classifier/regressor are input into a custom Python program tasked to make an executive decision regarding the chemical identity and nomenclature (i.e., cerebroside, saturated, glucosyl, carbon chain length 8 = $GlcCer_8$). Overall, our SERS-based chemical taxonomy can distinguish multiple key structural characteristics of unknown molecules, even down to the functional group levels. It can also provide accurate predictions of the molecules' complete molecular structure and identity. If we omit any of the RF-Cs or SVM-Rs, we lose the ability to determine a specific structural characteristic, precluding holistic structural elucidation (Fig. 4g, h). For instance, removing RF-C4 would leave the identity of the glycosyl moiety unidentified. Removing the SVM-Rs would mean we could not predict the cerebroside's ceramide carbon chain length. Notably, the predictions are instantaneous due to ML's ability to analyze chemically relevant spectral information quickly and accurately.

We demonstrate full generalizability of our SERS-based chemical taxonomy in forward predicting the identity of all 11 gluco- and galactocerebrosides using their SERS spectra. First, we systematically remove individual cerebroside sets of 60 spectra from the total 840 spectra (60 each for MPBA blanks, two monosaccharides, and 11 cerebroside samples at $10^{-4}$ M) for blind predictions (Table 1). To mitigate the risk of chance error, we randomly stratify the remaining 780 spectra into training and test using 5-fold cross-validation over 100 iterations (Suppl. Note 7, Suppl. Tables 2–12). Collectively, the four

**Table 2 | Forward prediction results of $GalCer_{16}$ at $10^{-5}$–$10^{-10}$ M used as blind tests, which are 1 – 6 orders of magnitude lower than the concentration used in the trained model ($10^{-4}$ M)**

| [GalCer16] (M) | Overall classification accuracy | Predicted carbon chain length | Absolute difference | Predicted Cerebroside |
|---|---|---|---|---|
| $10^{-5}$ | 100% | 16 ± 1 | 0 | Galcer16 |
| $10^{-6}$ | 98% | 16 ± 0 | 0 | Galcer16 |
| $10^{-7}$ | 93% | 16 ± 1 | 0 | Galcer16 |
| $10^{-8}$ | 93% | 17 ± 0 | 1 | Galcer16 |
| $10^{-9}$ | 97% | 16 ± 1 | 0 | Galcer16 |
| $10^{-10}$ | 87% | 15 ± 1 | 1 | Galcer16 |
| Average | 94 ± 5% | | 0.3 ± 0.5% | |

RF-Cs achieve >90% cumulative classification accuracy across all models and test instances, successfully determining various structural characteristics. The two SVM-Rs for GlcCer and GalCer also accurately predict the carbon chain length of epimeric cerebrosides, differing by at most one carbon, thus allowing for complete structural elucidation.

Next, we demonstrate that even when our chemical taxonomy is built using cerebrosides at $10^{-4}$ M concentrations, it can effectively predict the identity of "unidentified molecules" ranging from $10^{-5}$ to $10^{-10}$ M, which is 1–6 orders of magnitude lower than the trained model (Table 2). Such concentration-independence prediction capability is critical for practical SERS sensing applications where the concentration of analytes is often unknown. In this case, we input untrained $GalCer_{16}$ SERS measured at $10^{-5}$–$10^{-10}$ M into the chemical taxonomy built using the spectra of other epimers and blanks at $10^{-4}$ M. The chemical taxonomy returns with 87–100% classification accuracies, <1 carbon length difference for the entire concentration range (Table 2, Suppl. Note 7, Suppl. Tables 13–18). Critically, we achieve accurate prediction at such a dynamic range of concentrations because of the robust SERS spectra differences of each epimer-MPBA SERS fingerprint. These results are direct evidence of the generalizability of our model built upon sound chemical knowledge using chemically relevant features as inputs. Overall, our SERS-based chemical taxonomy ML framework enables an end-to-end structural elucidation and identification of 11 cerebrosides at any concentration not trained in the model through stepwise elucidation of their multifarious structural characteristics from their SERS fingerprints. This finding signifies a step forward for ML-driven SERS as a toolkit for instantaneous, untargeted chemical identification and differentiation of isomeric (bio)molecules that were previously an elusive class of epimer for SERS due to their small Raman cross-sections.

## Quantification and multiplex quantification

After developing a chemical taxonomy for the molecular structure elucidation and identification of cerebrosides, we proceed to construct separate SVM-R models for quantification. These models are designed to accurately quantify the concentrations of all 11 pure cerebrosides from $10^{-4}$ to $10^{-10}$ M. Our models show near-ideal linearity spanning seven orders of magnitude with $R^2$ of 0.95–1.00 and low $RMSE_{prediction}$ of 0.09-0.44 for each epimer, confirming the ultratrace sensitivity of our SERS platform with a detection limit of $10^{-10}$ M (Fig. 5a–l, Suppl. Note 5). In contrast, the complex derivatization procedure required for gold standard LC-MS analysis has typically hindered the accurate quantification of these biomolecules due to poor epimer recovery[2,3].

In addition, we further achieve multiplex quantification within the physiologically relevant $\mu$M range of the epimeric $GalCer_{24:1}$ and $GlcCer_{24:1}$, which coexist in the human body and are vital biomarkers for endometriosis and Gaucher as well as Fabry's and Krabbe diseases, respectively[14–18]. To construct a multiplex quantification model for binary mixtures of $GlcCer_{24:1}$ and $GalCer_{24:1}$, we first build a calibration curve by varying the mol% of $GalCer_{24:1}$ from 0 to 100% (and vice versa for $GlcCer_{24:1}$) using 60 SERS spectra for each calibration set. The total concentration is maintained at 100 $\mu$M (Fig. 5m, n, Suppl. Note 7, Suppl. Tables 19–21). Our calibration curve exhibits a near-ideal linearity with a cross-validation $R^2$ value of 0.99 and a low $RMSE_{calibration} = 2.18$, indicating good predictive accuracy (Fig. 5n). Composition predictions of three binary mixtures as blind tests comprising 90%, 60%, and 40% $GalCer_{24:1}$, respectively, also exhibit good linear coefficient $R^2 > 0.93$, $RMSE_{prediction} = 5.5$ and an absolute difference of <4 $\mu$M or 4% between the predicted and actual concentrations (Fig. 5m, n, Table 3). Importantly, we demonstrate excellent detection sensitivity even with minute changes in the concentrations of two cerebrosides, underpinning the potential of our SERS strategy to quantify them in biofluid mixtures concurrently.

Finally, we synergize the chemical taxonomy framework with these 11 cerebroside SVM-R quantification models to demonstrate simultaneous structural elucidation, molecular identification, and quantification of "unidentified cerebrosides" across concentrations ranging from $10^{-4}$ to $10^{-10}$ M (Tables 4, 5, Suppl. Note 7, Suppl. Tables 22–24). As a proof-of-concept, we test 30 blind samples of three different cerebrosides with various concentrations near the detection limits. We achieve predictive performance with >80% cumulative classification accuracies and <10% quantification errors. Our results demonstrate that the majority of spectra in each blind test can be correctly classified and identified. We note a slight decrease in classification accuracy when the samples are at or near the detection limit (LOD) of $10^{-10}$ M. For instance, when we test 10 blind samples of $GalCer_{12}$ at $10^{-8}$ M, one sample is wrongly classified as unsaturated, resulting in a drop in cumulative accuracy to 90%. Nevertheless, it is essential to highlight that by implementing a majority voting scheme, where the predictions are based on the results of most samples, we can confidently identify all cerebrosides, even at their LODs of $10^{-10}$ M. Overall, we perform quantitative and multiplex quantitative cerebroside detection at the physiologically relevant micromolar level with high predictive accuracies. In our quantitative detection system, we only require five $\mu$L of sample, and the whole procedure requires <1 h, including sample mixing and drying, SERS measurements, and ML predictions, which is significantly faster than conventional LC-MS analyses (h to days) for cerebroside profiling. This finding establishes our ML-driven SERS approach as a promising tool for ultra-trace quantitatively detecting large biomolecules with small Raman scattering cross-sections for biomedical applications.

In conclusion, the establishment of an integrated SERS-based chemical taxonomy ML framework demonstrates the capacity for predictive modeling, enabling untargeted structural elucidation and identification with >90% classification accuracies quantification of 11 epimeric cerebrosides at trace concentrations <10% error using their SERS spectra. We also provide an in-depth understanding of the spectral regions contributing to the differentiation of all 13 epimers by corroborating with DFT simulations through systematic investigation and profiling of their molecular structures and SERS spectral characteristics. Using the five-level hierarchical chemical taxonomy ML framework, (bio)molecules are sequentially classified according to confirmable differences and similarities in molecular structural characteristics and finally identified by piecing the collective information gained from each model. These structural characteristics include (1) blank vs. epimer (2) monosaccharides vs. cerebrosides, (3) saturated vs. unsaturated, (4) glucosyl vs. galactosyl, and (5) the exact carbon chain length of the ceramide. Our research underscores the high cumulative classification accuracy of >90% for the four RF-C and up to 1 carbon discrepancy when predicting the carbon chain length using the SVM-Rs. Moreover, the integrated ML pipeline enables quantitative detection of all 11 pure cerebrosides after identifying them from $10^{-4}$ to $10^{-10}$ M, showing good predictive accuracy and near-ideal linearity spanning seven orders of magnitude. We further demonstrate multiplex SERS quantification of 30 blind test epimer binary mixtures (total concentration of 100 $\mu$M) with <4% difference between the actual and predicted concentrations. Overall, our concentration-independent ML-driven SERS chemical taxonomy can forward-predict epimeric cerebrosides over a wide range concentration range of $10^{-4}$–$10^{-10}$ M and allows rapid, one-step untargeted structural elucidation and quantification of "unidentified" isomeric biomolecules. We envision the exploration of biomolecules characterized by higher degrees of saturation (>2) and the ability to elucidate further the exact location of the C = C bonds along the carbon chain and their cis-trans isomerism. The presence of multi-isomeric sites in complex diastereomers may also be probed. Lastly, to extend the framework untargeted elucidation of other classes of isomeric compounds beyond the 13 epimers used in this study, we posit the creation of a global SERS molecular space using high-throughput platforms to test various probe-analyte combinations. This innovation can synergize effectively with miniaturized SERS spectrometers and microfluidic chips to realize the point-of-need lab-on-a-chip concept by streamlining sample separation and pretreatment to improve SERS detection in complex and heterogeneous mediums.

## Methods

### Chemicals

Silver nitrate ($AgNO_3$, ≥99%), anhydrous 1,5-pentanediol (PD, ≥97%), poly(vinylpyrrolidone) (PVP, average $M_w = 55,000$ g $mol^{-1}$), 1H,1H,2H,2H-perfluorodecanethiol (PFDT, ≥97%), 4-mercaptophenylboronic acid (4-MPBA, ≥90%) and, dodecane ($C_{12}H_{26}$, anhydrous, ≥99%), D-(+)-galactose ($C_6H_{12}O_6$, ≥99%), D-(+)-glucose (dextrose $C_6H_{12}O_6$, ≥99.5%, GC) were purchased from Sigma Aldrich. Copper (II) chloride was purchased from Alfa Aesar. Glucosylceramides (GlcCer C8), GlcCer($\beta$) ceramide (d18:1/8:0 ≥ 99%); GlcCer C12, GlcCer($\beta$) ceramide (d18:1/12:0 ≥ 99%); GlcCer C16, GlcCer($\beta$) ceramide (d18:1/16:0 ≥ 99%); GlcCer C18, GlcCer($\beta$) ceramide (d18:1/18:0 ≥ 99%); GlcCer C24:1 GlcCer(ß) ceramide (d18:1/24:1(15Z) ≥ 99%) and galactosylceramide (GalCer C8), GalCer($\beta$) ceramide (d18:1/8:0 ≥ 99%); GalCer C12 GalCer($\beta$) ceramide (d18:1/12:0 ≥ 99%); GalCer C16 GalCer($\beta$) ceramide (d18:1/16:0 ≥ 99%); GalCer C18 GalCer($\beta$) ceramide (d18:1/18:0 ≥ 99%); GalCer C24 GalCer($\beta$) ceramide (d18:1/24:0 ≥ 99%); GalCer C24:1 GalCer(ß) ceramide (d18:1/24:1(15Z) ≥ 99%) were purchased from Avanti® Polar Lipids, Inc. Ethanol ($C_2H_6O$, ACS, ISO, Reag. Ph Eur), acetone ($C_3H_6O$, HPLC, ≥99.9%) and potassium hydroxide (KOH, ACS reagent, ≥85%, pellets) were obtained from Merck. Milli-Q water (>18.0 MΩ cm) was purified with a Sartorius Arium® 611 UV ultra-pure water system. All reagents were used without further purification.

### Synthesis and purification of silver nanocubes

Ag nanocubes were synthesized via the polyol method[31]. Two precursor solutions were first prepared. Precursor solution 1 consisted of

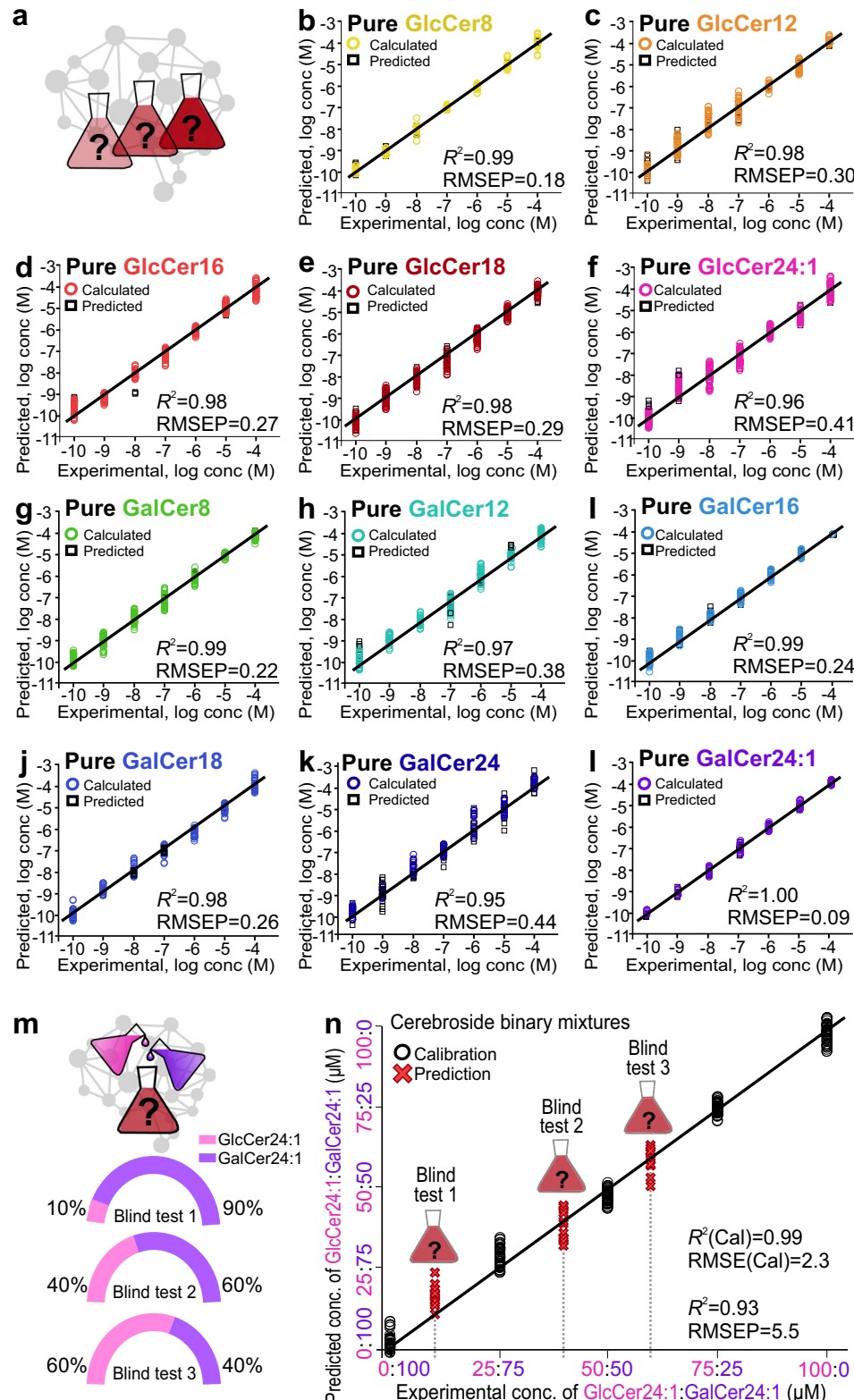

**Fig. 5 | Single and multiplex SERS quantification results. a** Schematics of the pure cerebrosides of different concentrations. SERS quantification of different pure **b**–**f** GlcCers and **g**–**l** GalCers from 10⁻⁴ to 10⁻¹⁰ M using SVM-R models. **m** Scheme of the three multiplex mixtures of epimeric GlcCer$_{24:1}$ and GalCer$_{24:1}$.

**n** Multiplex quantification of the three mixtures with different percentage compositions of the two cerebrosides constituting a total concentration of 100 μM using an SVM-R model.

**Table 3 | Multiplex quantification results for 3 sets of binary mixtures with total cerebroside concentration of 100 μM as blind tests consisting 10 individual samples for each set of mixture**

| | GlcCer24:1 | | | GalCer24:1 | | |
|---|---|---|---|---|---|---|
| | Actual concentration (μM) | Predicted concentration (μM) | Absolute difference (μM) | Actual concentration (μM) | Predicted concentration (μM) | Absolute difference (μM) |
| Blind test 1 (samples 1–10) | 10 | 14 ± 2 | 4 | 90 | 86 ± 2 | 4 |
| Blind test 2 (samples 11–20) | 40 | 39 ± 3 | 1 | 60 | 61 ± 3 | 1 |
| Blind test 3 (samples 21–30) | 60 | 58 ± 3 | 2 | 40 | 42 ± 3 | 2 |

silver nitrate (0.50 g) and copper (II) chloride (0.86 μg) dissolved in PD in a scintillation vial. Precursor solution B consisted of PVP (0.25 g) dissolved in PD. 20 mL of PD was added to a 100 mL round-bottom flask and heated at 190 °C for 10 min in a temperature-controlled silicon oil bath. Subsequently, aliquots of PVP (250 μL) and silver nitrate (500 μL) precursor solutions were injected in alternation to the reaction flask at different rates, namely 500 μL every min for silver nitrate and 250 μL every 30 s for the PVP solution, until the reaction mixture turned reddish-brown. The as-synthesized Ag nanocubes were purified via several rounds of centrifugation at 12,000 × $g$ and sonication in acetone and ethanol, then subsequently stored in ethanol. Ag nanocubes were further subjected to vacuum filtration using polyvinylidene fluoride filter membranes (Durapore®) with pore sizes 5 μm, 0.65 μm, 0.45 μm, and 0.22 μm to remove impurities before use.

### SEM and UV characterization of Ag nanocubes

The synthesized Ag NCs were subjected to scanning electron microscopy (SEM) using the JEOL JSM-7600F Schottky field emission electron microscope at an accelerating voltage of 5 kV. Measurements were randomly taken at 5 different spots on the SEM substrate to get a representative group of images for each Ag nanocube sample. For each sample of Ag NCs, the size (edge length) of 100 randomly selected nanocubes was measured using the ImageJ freeware. The UV-vis spectra were taken on the Agilent Technologies Cary 60 UV/visible spectrophotometer.

### Surface functionalization of Ag nanocubes

The purified and filtered Ag nanocubes underwent 2 rounds of ligand exchange with MPBA. Briefly, Ag nanocubes (ethanolic, 4 mg mL$^{-1}$, 400 μL) were added to ethanol (1600 μL) while stirring at 750 rpm for 1 min. Next, 4-MPBA (ethanolic, 1 mM, 200 μL) was injected and the mixture was stirred for 3 h at 750 rpm in the dark. After the first ligand exchange cycle of 3 h, the 4-MPBA functionalized Ag nanocubes were purified via two rounds of centrifugation at 12,000 × $g$ and sonication in ethanol to remove any excess 4-MPBA and redispersed in 400 μL of ethanol. The ligand exchange process was repeated but stirred for 2 h instead of 3 h in the second cycle. The 4-MPBA functionalized Ag nanocubes (ethanolic, 4 mg mL$^{-1}$) were stored at 4 °C.

### Analyte preparation and reaction

Glucose (aq, 5 mM), Galactose (aq, 5 mM), and 11 other cerebrosides of different chain lengths (98:2 ethanol: dodecane, 5 mM) stock solutions were prepared. Serial dilution was carried out to yield 7 concentrations per cerebroside (1 mM to 1 nM). 4-MPBA functionalized Ag nanocubes (ethanolic, 4 mg mL$^{-1}$) were sonicated and redispersed in pH 11 KOH solution, before reaction with the analytes, and immediate SERS measurements. 4-MPBA-Ag nanocubes (aqueous (aq.) pH 11, 4 mg mL$^{-1}$, 10 μL) were added to pH 11 KOH solution (aq., 35 μL) before adding the respective analytes (5 μL). The mixtures were sonicated immediately after each addition to that it is well-mixed and further shaken (Eppendorf ThermoMixer C, 1000 rpm at 25 °C) for 30 min to ensure thorough mixing during the reaction, including three sonication and vortex steps at the tenth, twentieth and thirtieth-min mark. After the reaction, 5 droplets of 2 μL of the mixture sample solution were drop-casted on different locations on the perfluorothiol Ag hydrophobic substrate and dried under ambient conditions. SERS spectra were then collected on the 5 dried spot areas using a laser beam with the conditions listed below.

### SERS measurement of 13 analytes

SERS measurements were performed using $x$-$y$ hyperspectral imaging modes of the Ramantouch microspectrometer (Nanophoton Inc., Osaka, Japan) with a 532 nm excitation laser (power = 0.10 mW). A 20× objective lens was used with 30 s acquisition time. The spectral window of 400–1800 cm$^{-1}$ was used for data analyses. The spectra were

 

**Table 4 | Results of our proof-of-concept simultaneous structural elucidation, molecular identification, and quantification of 3 cerebrosides (10 samples each) at various concentrations using our integrated SERS-based chemical taxonomy framework**

| | Molecular identity | | | Concentration | | |
|---|---|---|---|---|---|---|
| | Actual | Predicted | Accuracy (%) | Actual | Predicted | Difference (%) |
| Blind test 1 (samples 1–10) | GalCer12 | GalCer12 | 90 | $1 \times 10^{-8}$ M | $1.07 \times 10^{-8}$ M | 9 |
| Blind test 2 (samples 11–20) | GlcCer16 | GlcCer16 | 100 | $1 \times 10^{-6}$ M | $1.10 \times 10^{-6}$ M | 10 |
| Blind test 3 (samples 21–30) | GalCer24 | GalCer24:1 | 80 | $1 \times 10^{-10}$ M | $1.06 \times 10^{-10}$ M | 8 |

preprocessed using baseline correction via the adaptive iteratively reweighted penalized least squares (airPLS) algorithm and min-max normalization (max = 1). For each of the 13 analytes, 3 different spectra were measured from different locations within the 5 drop-casted droplets (total 15 spectra per analyte) and the experiment was repeated 4 times (total 60 spectra per analyte per concentration) to ensure reproducibility. Representative SERS spectra were obtained by averaging 60 individual SERS spectra per analyte per concentration and data analysis was completed using Origin 9.0 software (OriginLab Corporation, Northampton, MA, USA).

### Fabrication of the hydrophobic substrate
Oxygen plasma (FEMTO SCIENCE, CUTE-MP/R, 100 W) was used to clean and prepare the silica substrates for 5 min. Next, chromium (Cr) and silver (Ag) films were deposited in sequence using a home-built Syskey splutter system via thermal evaporation deposition. An adhesion layer of Cr (12.5 nm) was first deposited, followed by an Ag 100 nm film Si substrate. The deposition rates of Cr and Ag were 0.1 and 0.5 Å s$^{-1}$, respectively and the rate was monitored in situ using a quartz crystal microbalance. Cr and Ag targets (99.99% purity) were purchased from Advent Research Materials, UK. The resulting coated Ag nanocube array was then functionalized by immersing a 5 mM 1H,1H,2H,2H-perfluorodecanethiol ethanolic solution for at least 15 h before rinsing three times with ethanol to remove any unbounded PFDT and stored in nitrogen before use.

### Contact angle measurements
Static contact angles were measured on the Theta Lite fully automated optical tensiometer equipped with a Firewire digital camera and Attention from Biolin Scientific by drop-casting a sessile 4 μL water droplet onto the hydrophobic and Si substrates respectively. All contact angle measurements reported were repeated at least five times across each substrate and averaged.

### Determining the analytical enhancement factor (AEF)
The analytical enhancement factor is calculated by using the equation below:

$$AEF = I_{SERS}/I_{Raman} \times C_{Raman}/C_{SERS} \tag{1}$$

Where $I_{SERS}$ and $I_{Raman}$ were the intensities from the signals recorded on SERS and normal Raman, and $C_{SERS}$ and $C_{Raman}$ were the corresponding analyte concentrations measured using a hydrophobic platform and normal Si-wafer Raman substrate respectively. We choose the CC stretching ($a_1$, $\nu$(CC)) peak at 1598 cm$^{-1}$ of the MPBA normal Raman spectrum and the corresponding peak at 1575 cm$^{-1}$ of the 4-MPBA SERS spectrum. We conduct this experiment by using 2 μL of MPBA solutions at different concentrations. For the normal Raman measurements, we drop-casted a 4-MPBA solution (1 M, 2 μL) on Si wafer substrates. For the SERS measurements, we dropcast a 4-MPBA-Ag nanocube solution ($10^{-2}$ M, 2 μL) on the Si wafer substrate. For the SERS measurements on our hydrophobic substrate, we drop cast a 4-MPBA-Ag nanocube solution ($10^{-4}$ M, 2 μL) on the Si wafer substrate. All the SERS spectra in this work are collected under dry conditions

after the solvent in the droplet has fully evaporated, and the AEF was calculated based on the average intensities of the corresponding vibrational bands at 1575 cm$^{-1}$ and 1598 cm$^{-1}$ of MPBA in the 25 spectra for each substrate. SERS measurements were taken with the SERS conditions listed above.

### DFT simulations
The calculations on the interaction of the MPBA-functionalized Ag surface with all 13 analytes (2 monosaccharides and 11 cerebrosides) were carried out using the unrestricted B3LYP exchange–correlation function in the Gaussian 09 computational chemistry package. The 6-31G (d,p) basis set was used for C, H, O, B, and N. The LANL2DZ basis set was employed for Ag. The Ag surface was modeled using a reported triangle consisting of six Ag atoms. Structure optimization was carried out in 3 steps. Firstly, we optimized the geometry of all 13 analytes. Secondly, we optimized the geometry of the triangular Ag cluster; then 4-MPBA was placed on its vertex, and the whole system was reoptimized. Lastly, we introduced the optimized analyte molecules to the MPBA-Ag system and formed 2 boric ester bonds with MPBA via the $C_3$ and $C_4$ hydroxy groups, and the whole system was reoptimized with Ag atoms fixed again.

### Chemometrics analysis
Unsupervised principal component analysis (PCA) was performed using SOLO v8.8 (Stand Alone Chemometrics Software, Eigenvector Research, Inc.). The PCA model was cross-validated using Venetian blinds, with 10 splits and a blind thickness of 1. Supervised support vector machine regression (SVM-R) with prior PCA compression applied to prevent overfitting, and extreme gradient boosting classification where eta/learning rate = 0.1, max depth = 6, num_round = 500 was performed using SOLO v8.8 (Stand Alone Chemometrics Software, Eigenvector Research, Inc.). (A) For the forward prediction model (regression model 5), the training dataset containing all chain lengths except the testing dataset (e.g., train = GlcCer$_{8,16,18 \text{ and } 24}$ and test = GlcCer$_{16}$) was first randomly stratified into an 80% train and 20% cross-validation for model training/building. Once trained, the test dataset was used to assess the model accuracy by determining the $R^2$ value, root mean square error of prediction (RMSEP), predicted chain length, and% difference between the predicted value and actual value. (B) For the pure analyte regression, the dataset was first randomly stratified into a 75% train and 25% test in one iteration. The test set was used to assess the model accuracy by determining the $R^2$ value, root mean square error of prediction (RMSEP). (C) For the multiplex regression model, the training dataset containing all training concentrations (e.g., GlcCer$_{24:1}$: GalCer$_{24:1}$ ratios are 0:100, 25:75, 50:50, 75:25, 100:0) was first randomly stratified into 80% train and 20% cross-validation for model training/building. Once trained, the 3 testing datasets (GlcCer24:1: GalCer24:1 ratios are 10:90, 40:60, 60:40) were used to assess the model accuracy by determining the $R^2$ value, root mean square error of prediction (RMSEP), predicted chain length, and difference in percent between the predicted and actual value.

All other chemometrics analyses including unsupervised t-distributed stochastic neighbor embedding (t-SNE) and PCA as well as the supervised random forest, decision tree, support vector

**Table 5 | A detailed breakdown of the ML results obtained by our integrated SERS-based chemical taxonomy framework using 10 blind test samples of GalCer$_{12}$ with the actual concentration of 1 × 10$^{-8}$ M. The probability scores for each sample is recorded below**

| ML model | RF-CI | | RF-C 2 | | RF-C 3 | | RF-C4 | | SVM-R 5.1 & 5.2 Carbon chain length | | GalCer12 quantification | | |
| --- | --- | --- | --- | --- | --- | --- | --- | --- | --- | --- | --- | --- | --- |
| Samples | Epimer | Blank | Cerebroside | Monosaccharide | Saturated | Unsaturated | GlcCer | GalCer | Predicted carbon chain length | Absolute difference | Predicted conc. (nM) | Difference (nM) | Difference (%) |
| | 1 | 0 | 0.7 | 0.3 | 0.51 | 0.49 | 0.26 | 0.74 | 12.2 | 0.2 | 12 | 2 | 20 |
| | 1 | 0 | 0.7 | 0.3 | 0.61 | 0.39 | 0.35 | 0.65 | 12.5 | 0.5 | 10 | 0 | 1 |
| | 1 | 0 | 0.77 | 0.23 | 0.61 | 0.39 | 0.26 | 0.74 | 12.5 | 0.5 | 9.4 | 0.6 | 6 |
| | 1 | 0 | 0.8 | 0.2 | 0.48 | 0.52* | 0.38 | 0.62 | 12.5 | 0.5 | 9.5 | 0.5 | 5 |
| | 1 | 0 | 0.8 | 0.2 | 0.6 | 0.4 | 0.33 | 0.67 | 12.5 | 0.5 | 10.2 | 0.2 | 2 |
| | 1 | 0 | 0.76 | 0.24 | 0.6 | 0.4 | 0.26 | 0.74 | 12.4 | 0.4 | 10.1 | 0.1 | 1 |
| | 1 | 0 | 0.7 | 0.3 | 0.52 | 0.48 | 0.38 | 0.62 | 12.5 | 0.5 | 11 | 1 | 10 |
| | 1 | 0 | 0.77 | 0.23 | 0.6 | 0.4 | 0.34 | 0.66 | 12.5 | 0.5 | 11.2 | 1.2 | 12 |
| | 1 | 0 | 0.80 | 0.2 | 0.61 | 0.39 | 0.26 | 0.74 | 13.8 | 1.8 | 11.5 | 1.5 | 15 |
| | 1 | 0 | 0.70 | 0.3 | 0.51 | 0.49 | 0.26 | 0.74 | 12.4 | 0.4 | 12 | 2 | 20 |
| Average | 100% | | 100% | | 90% | | | 100% | 12.6 ± 0.4 | | 10.7 ± 0.9 | | 9 |

*Denotes wrong prediction where the probability of the wrong class is > 0.50.

machine, Naïve Bayesian network, and neural network classification models were conducted using Orange Data Mining[32]. The following parameters were applied for each model: t-SNE using perplexity = 30, Exaggeration = 1, PCA components = 15; random forest where the number of trees = 1000, number of attributes arbitrarily considered at each split = $\sqrt{}$(number of attributes), Max depth = 10, No splitting of subsets <5; decision tree where the number of trees = 50, nodes = 2; support vector machine, cost = 1, loss = 0, 10 RBF kernel; naïve Bayesian network using, cost = 1, loss = 0, 10 RBF kernel, neural network using 50 nodes per layer, 2 layers. For all classification models, where all classes are represented, the dataset was first randomly stratified into a 75% train and 25% test in one iteration. The test set was used to assess the model accuracy by determining its prediction accuracy and F1 score. This process was repeated for another 99 iterations to derive the average prediction accuracy for the class. For the RF classification models used in forward prediction (classification models 1–4), the dataset containing all classes, except the blind test class, was first randomly stratified into an 80% train and 20% cross-validation for 100 iterations to derive the average prediction accuracy for the model. Once trained, Once trained, the test datasets were used to assess the 4 different models' accuracy by assessing the classification accuracy, Precision, Recall, and F1 score.

Machine learning metrics were calculated as follows:

Classification accuracy
$$= \frac{\text{True positive} + \text{True negative}}{\text{True positive} + \text{False positive} + \text{True negative} + \text{False negative}} \quad (2)$$

$$\text{Precision} = \frac{\text{True positive}}{\text{True positive} + \text{False positive}} \quad (3)$$

$$\text{Recall} = \frac{\text{True positive}}{\text{True positive} + \text{False negative}} \quad (4)$$

$$\text{F1score} = \frac{2 \times \text{Precision} \times \text{Recall}}{\text{Precision} + \text{Recall}} \quad (5)$$

$$\text{Difference} = \frac{|\text{Experimental value} - \text{Predicted value}|}{\text{Experimental value}} \quad (6)$$

## Data availability
The data that support the findings of this study are available via Zenodo[33] and from the corresponding authors upon request.

## Code availability
Codes are available via Zenodo[33] and from the corresponding authors upon request.

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

## Acknowledgements
This research is supported by Singapore National Research Founda-tion Central Gap Fund (NRF2020NRF-CG001-010) X.Y.L., Competitive Research Programme (NRF-CRP26-2021-0002) X.Y.L, National Research Foundation Investigatorship (NRF-NRFI08-2022-0011) X.Y.L, A*STAR AME Individual Research Grant (A20E5c0082) X.Y.L and Institute for Digital Molecular Analytics and Science (IDMxS) under Research Centres of Excellence Scheme, Singapore Ministry of Education X.Y.L. National Research Foundation, Prime Minister's Office, Singapore under its Campus for Research Excellence and Technological Enterprise (CREATE) programme, through Singapore MIT Alliance for Research and Technology (SMART): Critical Analytics for Manufacturing Personalised-Medicine (CAMP) Inter-Disciplinary Research Group. Y.H.L.

## Author contributions
X.Y.L., E.X.T., and Y.H.L. conceptualized the study. E.X.T. planned the methodology; E.X.T., W.A.L., and J.Y.N. carried out the experimental studies. E.X.T. conducted the formal analysis. E.X.T. performed the computational studies and machine learning modeling. E.X.T. wrote the original draft and drew the figures. E.X.T., X.Y.L., S.X.L., I.Y.P., Y.H.L., and N.S.T. reviewed and edited the manuscript. X.Y.L. supervised the work. X.Y.L. and Y.H.L. acquired funding for the work.

## Competing interests
The authors declare no competing interests.
