## [Peer Review File · Nature Communications]

Forward-predictive SERS-based chemical taxonomy for untargeted structure elucidation of epimeric cerebrosidesREVIEWER COMMENTS

Reviewer #1 (Remarks to the Author):

Taking the 11 epimeric cerebroside as example, the authors reported an integrated SERS-based chemical taxonomy machine learning framework for untargeted structural elucidation. They firstly divide all normalized and baselined SERS spectra into 6 regions and then fit the peaks to generate 19×5 features. Then they developed random forest models and support vector machine regressors to achieve the structural characteristic: presence/absence of epimers, monosaccharide/cerebroside, saturated/unsaturated cerebroside, glucosyl/galactosyl, and GlcCer or GalCer's carbon chain lengths. The Clear t-sne and the accurate prediction of the carbon chain differences show its high accuracy. My concerns are shown as follows:

1. The manuscript does not provide the code and data of the machine learning framework, which is disadvantageous for readers who want to verify and use the framework.
2. The elucidation of the epimer-specific SERS fingerprints on Line 173-234 and Figure 2 is very clear. Have the authors used pure mathematical tools, such as MATLAB, to analyse the structural characteristics.
3. The manuscript should add a comparative experiment to try several machine learning methods (such as Bayesian, k-Nearest Neighbors and neural networks), and compare their accuracy with random forest classification and SVM regression. This may improve the accuracy of the framework using the best of them.
4. Figure 4A shows the SERS quantification error is about 2 order. For example, the 10^{-7} M to 10^{-9} M concentration of GlcCer12 GlcCer18 GalCer18 GalCer24 may be not distinguished. Have the authors considered to further optimize the ML models.
5. The authors should try using the whole spectra as the features to establish the ML framework.

Reviewer #2 (Remarks to the Author):

This is an excellent article that pushes the boundary of chemical identification using SERS and ML. The paper is really well written and has a high depth of information. I particularly like the sequential levels of identification using the RF algorithm THIS is a clever idea and works really well. In addition, the performance of the sensor for these cerebroside is well characterized. This is a really complete paper that is nearly ready for publication. I only have minor comments to be addressed.

The introduction mentions that manual referencing is limited to known molecules existing in the database. A comment on the potential limitation of the current method is warranted. The method works well for a class of molecules that are structurally similar, but surely fails for different classes of molecules. This distinction is important.

On line 194, a difference of 2 cm^{-1} is observed, but is that statistically significant. What is the resolution of the Raman spectrophotometer?

Related to the first comment, are there molecules that could cause interferences in the predictive model?

line 371, $10(-10)\text{M}$ does not have the exponent.

Reviewer #3 (Remarks to the Author):

The authors provided a very detailed report on a SERS-based chemical taxonomy machine learning framework for structural elucidation of 11 epimeric cerebrosides, and obtained a high accuracy (>90%) and single epimer and multiplex quantification with errors <10%. Although it could be new for structural elucidation of epimeric cerebrosides, the methodology [especially the PCA/SVM-based machine learning (or mode identification technology)] is not new completely, see such as

- (i) Li, X.; et al, Detecting Esophageal Cancer Using Surface-Enhanced Raman Spectroscopy (SERS) of Serum Coupled with Hierarchical Cluster Analysis and Principal Component Analysis. *Applied spectroscopy* 69, 1334–1341 (2015).
- (ii) Zou, Sumeng; et al, Semi-quantitative analysis of multiple chemical mixtures in solution at trace level by surface-enhanced Raman Scattering, *Scientific Reports*, 2017, 7 : 0-6186.
- (iii) Zhao Ling-yi; et al, "SERS Detection and Efficient Identification of Heroin and Its Metabolites Based on Au/SiO₂ Composite Nanosphere Array", *Spectroscopy and Spectral Analysis*, 2023, 43 (10), 3150.

Additional Comments:

- (1) How about the suitability of the SERS chemical taxonomy framework, what are its limitations?
- (2) How about anti-interference? How about the reproducibility of the results if solution droplet contains some interfering agents, or solution temperature is changed, which could influence the signal intensity of analytes.
- (3) The authors claimed that the SERS-based chemical taxonomy is of full generalizability. They should present the corresponding results for some other analytes
- (4) The authors should give a comparison with the normal methods.
- (5) The manuscript is not concise enough and was written in a verbose and tedious way. Especially, the final paragraph in Introduction is too verbose.

Reviewer 1

We thank the reviewer for the positive appraisal and greatly appreciate his/her suggestions. In this revision, we have done the following:

- 1) Provided all data and codes used in this work.
- 2) Clarified our methodology for objective spectral analysis in Figure 2.
- 3) Supplemented additional information on the choice of ML models used.
- 4) Supplemented additional discussion on optimization of SVM regressors used for quantification.
- 5) Supplemented additional information on the choice of ML inputs used.

1. The manuscript does not provide the code and data of the machine learning framework, which is disadvantageous for readers who want to verify and use the framework.

We thank the reviewer for the suggestion. All raw data and codes, including (1) spectra preprocessing code, (2) stochastic SERS peak deconvolution, (3) automated boxplot, (4) statistical T and U tests, (5) sequential ML framework using orange software with demo test data, are submitted together with the revised manuscript and available for all to use.

2. The elucidation of the epimer-specific SERS fingerprints on Lines 173-234 and Figure 2 is very clear. Have the authors used pure mathematical tools, such as MATLAB, to analyse the structural characteristics.

We thank the reviewer for the question. We did not use MATLAB to analyze the data. Instead, we used Python to perform in-depth and systematic statistical analysis. Three Python codes were used to objectively automate the 1) deconvolution and parameterization of the baselined and normalized spectra to obtain the $19 \times 5 = 95$ features/parameters and 2) generation of the boxplots comparing the 19×5 features. 3) statistical T and U test of the 95 features/parameters for each analyte. These codes are submitted together with the revised manuscript and are available for all to use.

3. The manuscript should add a comparative experiment to try several machine learning methods (such as Bayesian, k-Nearest Neighbors and neural networks), and compare their accuracy with random forest classification and SVM regression. This may improve the accuracy of the framework using the best of them.

We thank the reviewer for the suggestion. We have tried several ML models for our proposed strategy, including the naïve Bayesian, neural network, support vector machines, extreme gradient boosting, and random forest models. After comparing their performance in classifying all 13 biomolecules, we selected the best-performing random forest classification and SVM regression. We have included those experiments in SI7, **Figure S8A-C**.

(A) ML model exploration for classification

(i) Decision tree

(ii) Naive Bayesian network

(iii) Support vector machine

(iv) Random forest

(v) Extreme gradient boosting

(vi) Neural network

(B) ML model classification results

Model	Classification accuracy	F1 score	Precision	Recall
Decision tree	0.985	0.985	0.985	0.985
Naive bayesian network	0.985	0.985	0.986	0.985
Support vector machine	0.962	0.961	0.963	0.962
Random forest	1.000	1.000	1.000	1.000
Extreme gradient boosting	0.990	0.990	0.990	0.990
Neural network	0.994	0.994	0.994	0.994

(C) Top 3 model performances over 100 iterations

(D) Comparison of model input (random forest)

Figure S8. Comparison of different ML models for classification of all 13 epimeric biomolecules. (A) confusion matrices of (i) decision tree, (ii) naïve Bayesian network, (iii) support vector machine (iv) random forest (v) extreme gradient boosting, and (vi) neural network ML models. (B) Compiled classification results. (C) comparison of the top 3 models' performance over 100 iterations. (D) Comparison of model inputs used for the best-performing random forest model.

4. Figure 4A shows the SERS quantification error is about 2 order. For example, the 10⁻⁷ M to 10⁻⁹ M concentration of GlcCer12, GlcCer18, GalCer18, GalCer24 may be not distinguished. Have the authors considered to further optimize the ML models.

We thank the reviewer for the suggestion. We have re-run the quantification SVM regressors on the Solo eigenvector software for cerebrosides with R^2 (prediction) < 0.98, specifically GlcCer12, GlcCer 24:1, GalCer12, and GalCer24. Specifically, we perform principle component analysis (PCA) data compression to the X-block data before calculating or applying the SVM model. Using a standard PCA model to compress the information makes the SVM more stable and less prone to overfitting. This improves the prediction results, especially in the testing/validation dataset for GlcCer12, GlcCer 24:1, and GalCer12. These improvements are reflected in **Figure 4A (ii, v, vii, and x)**.

Cerebroside	Original model		Optimized (re-run) model	
	RMSEP	R^2 (prediction)	RMSEP	R^2 (prediction)
GlcCer12	0.43	0.96	0.30	0.98
GlcCer24:1	0.46	0.95	0.41	0.96
GalCer12	0.44	0.96	0.38	0.97
GalCer24	0.44	0.95	0.44	0.95

Overall, we achieved excellent R^2 (prediction) ≥ 0.95 and good RMSEP of ≤ 0.45 . Considering the scale (-4 to -10) used in the model, an RMSEP ≤ 0.45 indicates $\sim 4\%$ difference between actual and predicted values, showing good predictive accuracy of our constructed quantification models.

5. The authors should try using the whole spectra as the features to establish the ML framework. We thank the reviewer for the suggestion. Indeed, we have tried using the whole spectra as ML input and noted two main disadvantages 1) slightly lower accuracies and, 2) longer model training time (higher computational costs) due to the larger dataset of features causing the curse of dimensionality- where accuracies fall when the number of feature greatly outweighs the number of samples. (Nat Methods, 15(6), 399-400) We have included those experiments in **SI7, Figure S8D**.

Reviewer 2

We thank the reviewer for the affirmative appraisal and greatly appreciate his/her suggestions. In this revision, we have done the following:

- 1) Discussed potential limitations of the current manual referencing methods in the introduction and possible extension of the framework in the introduction and conclusion.
 - 2) Clarified the spectral resolution of the Raman spectrometer used.
 - 3) Clarified the strategic choice of 4-MPBA as our probe as well as glucose and galactose as primary interfering molecules in this work.
 - 4) Included the exponent and checked the units.
1. The introduction mentions that manual referencing is limited to known molecules existing in the database. A comment on the potential limitation of the current method is warranted. The method works well for a class of molecules that are structurally similar, but surely fails for different classes of molecules. This distinction is important.

We thank the reviewer for the suggestion. We have included potential limitations of the current method of manual referencing to known molecules in existing databases in **the introduction, paragraph 1, third sentence**: “Currently, identifying known chemical molecules through manual referencing of existing SERS databases and literature is relatively straightforward. However, manually matching SERS peaks to vibrational modes is tedious and error-prone, especially when handling complicated spectra and large datasets.”

We have also made the distinction in the manuscript and discussed the possible extension of the framework and our molecular space to predict other classes of isomers in the **introduction, paragraph 4, last two sentences** “In this work, we create a localized SERS molecular space, within which our ML framework can both interpolatively and extrapolatively predict the 11 gluco and galactocerebrosides. We envision high-throughput testing of more probes and analytes combinations to further extend the framework and create a global SERS molecular space for untargeted elucidation of other classes of isomeric compounds to meet escalating demands for rapid, point-of-need analytical tools” as well as in **conclusion, last sentence** “Lastly, to extend the framework untargeted elucidation of other classes of isomeric compounds beyond the 13 epimers used in this study, we posit the creation of a global SERS molecular space using high-throughput platforms to test various probe-analyte combinations.”

2. On line 194, a difference of 2 cm^{-1} is observed, but is that statistically significant. What is the resolution of the Raman spectrophotometer?

We thank the reviewer for the question. The spectrometer used is the RAMANtouch by Nanophoton, Japan, which has a spectral resolution of 1.2 cm^{-1} (lower than the observed difference of 2 cm^{-1}). Additionally, the wavenumbers of each SERS spectrum were carefully corrected using the 520 cm^{-1} reference peak of a Si wafer to ensure that the observed peak shifts are significant and not due to instrumentation and/or calibration error, etc.

3. Related to the first comment, are there molecules that could cause interferences in the predictive model?

We thank the reviewer for the question. We use 4-mercaptopboronic acid (4-MPBA) as a molecular probe to increase platform specificity by covalently capturing the epimers. We recognize that molecules with 1,2 diols can also bind to MPBA and can be possible interferences. In this study, we identified glucose and galactose as primary interferences since they have the exact same hexose and are expected to bind to MPBA in similar orientations/configurations to gluco- and galactocerebrosides. These primary interferences can be effectively differentiated from each other and all 11 cerebrosides. Other more structurally different glycans such as mannose and lactose, are also expected to induce distinctive spectral change due to their specific yet differential binding orientations with MPBA and can be readily differentiated, as demonstrated in past literature: *Advanced Sensor Research*, 2300052.; *Talanta*, 165, 516-521.; *ACS nano*, 14(2), 2542-2552.; *Spectrochimica Acta Part A: Molecular and Biomolecular Spectroscopy*, 288, 122179.

4. line 371, $10(-10)\text{M}$ does not have the exponent.

We thank the reviewer for the comment. We have added the exponent and checked all units in the revised manuscript.

Reviewer 3

We thank the reviewer for the positive feedback and greatly appreciate his/her suggestions. In this revision, we have done the following:

- 1) Clarified the difference in methods, objectives, and achievements in our work compared to other SERS-ML work.
 - 2) Discussed limitations and possible extension of our framework in the introduction and conclusion.
 - 3) Clarified possible interferences and affirmed the reproducibility and reliability of our results.
 - 4) Clarified our strategic choice of glucose and galactose as primary interferences.
 - 5) Clarified the difference in our work compared to current SERS-ML and generic SERS work.
 - 6) Concise the manuscript wherever possible, especially the final paragraph of the introduction.
1. Although it could be new for structural elucidation of epimeric cerebroside, the methodology [especially the PCA/SVM-based machine learning (or mode identification technology)] is not new completely, see such as
- (i) Li, X.; et al, Detecting Esophageal Cancer Using Surface-Enhanced Raman Spectroscopy(SERS) of Serum Coupled with Hierarchical Cluster Analysis and Principal Component Analysis. Applied spectroscopy 69, 1334–1341 (2015).
 - (ii) Zou, Sumeng; et al, Semi-quantitative analysis of multiple chemical mixtures in solution at trace level by surface-enhanced Raman Scattering, Scientific Reports, 2017, 7 : 0-6186.
 - (iii) Zhao Ling-yi; et al, “SERS Detection and Efficient Identification of Heroin and Its Metabolites Based on Au/SiO₂ Composite Nanosphere Array”, Spectroscopy and Spectral Analysis, 2023, 43 (10), 3150.

We thank for reviewer for the comment. In the first highlighted work, the authors used unsupervised methods to differentiate/classify healthy individuals from pre- and post-operative esophageal cancer patients. HCA achieved 77.6 % accuracy and 65.2 % specificity, whereas PCA achieved 89.6 % accuracy and 91.3% specificity. The authors in the second work developed a PCA-based method to quantify binary, ternary solutions of thiolated probes at 10⁻⁶ M using SERS. Lastly, in the third work, the authors used PCA, HCA, and SVM to differentiate/classify pure heroin, 6-MAM, and morphine, achieving good classification accuracy up to 10⁻⁴ mg·mL⁻¹.

Although we used some common ML tools (PCA and SVM), the way we use them, our objectives, and our achievements are completely different. Beyond showing that we can differentiate/classify the 13 epimers (10⁻¹⁰ M) using a single model, which is common in current state-of-the-art ML-SERS work, we further highlight in our work, the creation of a ML framework consisting of an

ensemble of sequential models to elucidate the molecular structure of unknown molecules in a step-wise manner using just their SERS spectra. This forward predictive ability, which can identify unknowns outside of the training data/models, has not been demonstrated before. We clarified and highlighted our novelty in the **introduction, second paragraph, and last 3 sentences**: “Crucially, this approach facilitates unprecedented forward prediction, allowing for the deduction of “unknown” molecules situated beyond the boundaries of the ML model. Specifically, our proposed process systematically excludes alternative structural possibilities when the SERS spectra traverse the hierarchical levels of the chemical taxonomy, culminating in the precise identification of the exact molecular structure. In contrast, such forward prediction remains elusive through a single classification ML model, which inaccurately classifies the "unknown" molecule as one of the pre-existing labeled classes in that model.”

To illustrate and highlight our distinction with current ML-SERS work, we have included a scheme below:

2. How about the suitability of the SERS chemical taxonomy framework, what is its limitations?

We thank the reviewer for the question. The dataset of 13 molecules (780 spectra) allows us to create a localized molecular space, within which our ML framework can both interpolatively and extrapolatively predict their molecular structure, even if their spectra are not trained in the model before. We also discussed the possible extension of the framework and our SERS molecular space to predict other classes of isomers in the **introduction, paragraph 4, last two sentences**: “In this

work, we create a localized SERS molecular space, within which our ML framework can both interpolatively and extrapolatively predict the 11 gluco and galactocerebrosides. We envision high-throughput testing of more probes and analytes combinations to further extend the framework and create a global SERS molecular space for untargeted elucidation of other classes of isomeric compounds to meet escalating demands for rapid, point-of-need analytical tools” as well as in **conclusion, last sentence:** “Lastly, to extend the framework untargeted elucidation of other classes of isomeric compounds beyond the 13 epimers used in this study, we posit the creation of a global SERS molecular space using high-throughput platforms to test various probe-analyte combinations.”

3. How about anti-interference? How about the reproducibility of the results if solution droplet contains some interfering agents, or solution temperature is changed, which could influence the signal intensity of analytes.

We thank the reviewer for the question. When testing the SERS platform, we identified glucose and galactose as primary interferences as they contain the exact hexoses and are expected to bind to MPBA the same way as our targets gluco- and galactocerebrosides. We show in our work that they can be effectively differentiated from all 11 cerebrosides and each other. We also recognize other molecules that contain 1,2 diols that can bind to our MPBA probes, such as mannose and lactose. However, since these molecules’ structure and atomic arrangements are more different than glucose and galactose, they are expected to induce more distinctive spectral changes due to their differential binding orientations with MPBA and can be readily differentiated, as demonstrated in past literature, including *Advanced Sensor Research*, 2300052.; *Talanta*, 165, 516-521.; *ACS nano*, 14(2), 2542-2552.; *Spectrochimica Acta Part A: Molecular and Biomolecular Spectroscopy*, 288, 122179.

To ensure reproducibility, we systematically tested 75 separate SERS substrates and included the full platform characterization in SII. We did not try adding interfering agents, such as hydrophobic solvents, salts etc into the solution droplet tests if they would affect the drying or signal intensity on our hydrophobic SERS platform. Adding such interfering agents can be a future work dedicated to testing the hydrophobic platform.

Furthermore, all SERS measurements are conducted after the droplet dries down and thus are not susceptible to temperature-dependent intensity variations such as those pertaining to solution-based SERS measurements. Moreover, in our proposed strategy, the analyte-induced spectral differences are in terms of relative peak intensity ratios and peak positions, which are independent of the absolute signal intensities. To enable accurate comparisons across different SERS spectra

and avoid data misinterpretation due to variations in absolute signal intensities, min-max normalization was applied as a post-measurement preprocessing step to each SERS spectrum. The wavenumbers of each SERS spectrum were also corrected using the 520 cm⁻¹ reference peak of a Si wafer to ensure that the observed peak shifts are not due to instrumentation error, etc.

Overall, we would like to emphasize that the main focus of this work is primarily to showcase the use of our sequential ML framework to forward predict the molecular structure of unknown molecules in standardized and reproducible conditions. We have successfully demonstrated this concept using experimental data and theoretical simulations, whereby epimer-specific complex geometries are crucial to ensure high spectral specificity among different epimers (11 cerebroside and 2 monosaccharides).

4. The authors claimed that the SERS-based chemical taxonomy is of full generalizability. They should present the corresponding results for some other analytes.

We thank for reviewer for the suggestion. To avoid confusion, we have made the distinction in the manuscript in the **introduction, paragraph 4, sentence 8**: “highlighting the generalizability of our framework for structural elucidation of all 11 cerebroside”. Similar to the previous question, we identified glucose and galactose as the primary interferences that bind the most similarly to the gluco- and galactocerebroside and showed that they can be readily differentiated from all 11 target cerebroside. Their differentiation with other 1,2 diol-containing molecules, particularly glycans, can be seen in past literature (Advanced Sensor Research, 2300052.; Talanta, 165, 516-521.; ACS nano, 14(2), 2542-2552.; Spectrochimica Acta Part A: Molecular and Biomolecular Spectroscopy, 288, 122179.)

5. The authors should give a comparison with the normal methods.

We thank for reviewer for the suggestion. On one hand, similar to the reviewer’s first question/comment, we highlight the difference between our work and normal SERS-ML methods. On the other hand, if the normal method refers to the manual referencing of SERS spectra to known molecules in existing databases, we have also included a comparison in **the introduction, paragraph 1, third sentence**: “Currently, identifying known chemical molecules through manual referencing of existing SERS databases and literature is relatively straightforward. However, manually matching SERS peaks to vibrational modes is tedious and error-prone, especially when handling complicated spectra and large datasets.”

6. The manuscript is not concise enough and was written in verbose and tedious way. Especially, the final paragraph in Introduction is too verbose.

We thank the reviewer for the comment. We have revised the manuscript, especially the final paragraph in the introduction accordingly.

REVIEWERS' COMMENTS

Reviewer #1 (Remarks to the Author):

The current version is acceptable without any change.

Reviewer #1 (Remarks on code availability):

I have checked the code. The code is reproducible.

Reviewer #2 (Remarks to the Author):

The authors addressed all comments and the paper is now ready for publication in my opinion.

Reviewer #3 (Remarks to the Author):

The authors have addressed all issues the reviewer concerned. The manuscript in present version is much improved and could be acceptable.